# Effect of Substituents on Molecular Reactivity during Lignin Oxidation by Chlorine Dioxide: A Density Functional Theory Study

**DOI:** 10.3390/ijms241411809

**Published:** 2023-07-22

**Authors:** Baojie Liu, Lu Liu, Xin Qin, Yi Liu, Rui Yang, Xiaorong Mo, Chengrong Qin, Chen Liang, Shuangquan Yao

**Affiliations:** Guangxi Key Laboratory of Clean Pulp & Papermaking and Pollution Control, School of Light Industrial and Food Engineering, Guangxi University, Nanning 530004, China; liubaojie@st.gxu.edu.cn (B.L.); gxdxll123456@163.com (L.L.); 2105170239@st.gxu.edu.cn (X.Q.); yiliu@st.gxu.edu.cn (Y.L.); 2216391062@st.gxu.edu.cn (R.Y.); 2216301036@st.gxu.edu.cn (X.M.); liangchen@st.gxu.edu.cn (C.L.)

**Keywords:** chlorine dioxide, lignin model compound, density functional theory (DFT), substitute groups

## Abstract

Lignin is a polymer with a complex structure. It is widely present in lignocellulosic biomass, and it has a variety of functional group substituents and linkage forms. Especially during the oxidation reaction, the positioning effect of the different substituents of the benzene ring leads to differences in lignin reactivity. The position of the benzene ring branched chain with respect to methoxy is important. The study of the effect of benzene substituents on the oxidation reaction’s activity is still an unfinished task. In this study, density functional theory (DFT) and the m062x/6-311+g (d) basis set were used. Differences in the processes of phenolic oxygen intermediates formed by phenolic lignin structures (with different substituents) with chlorine dioxide during the chlorine dioxide reaction were investigated. Six phenolic lignin model species with different structures were selected. Bond energies, electrostatic potentials, atomic charges, Fukui functions and double descriptors of lignin model substances and reaction energy barriers are compared. The effects of benzene ring branched chains and methoxy on the mechanism of chlorine dioxide oxidation of lignin were revealed systematically. The results showed that the substituents with shorter branched chains and strong electron-absorbing ability were more stable. Lignin is not easily susceptible to the effects of chlorine dioxide. The substituents with longer branched chains have a significant effect on the flow of electron clouds. The results demonstrate that chlorine dioxide can affect the electron arrangement around the molecule, which directly affects the electrophilic activity of the molecule. The electron-absorbing effect of methoxy leads to a low dissociation energy of the phenolic hydroxyl group. Electrophilic reagents are more likely to attack this reaction site. In addition, the stabilizing effect of methoxy on the molecular structure of lignin was also found.

## 1. Introduction

As a free radical oxidizer, chlorine dioxide (ClO_2_) has good disinfection and sterilization properties [1,2]. It is widely used in production life and medical applications [3,4]. The excellent oxidation effect of chlorine dioxide makes it an efficient delignification reagent in elemental chlorine-free (ECF) bleaching processes [5]. During pulp bleaching, it can be taken up with the hydrogen (H) atom in the phenolic lignin structure according to a single electron transfer mechanism, and subsequently free radicals are formed. Since this reaction occurs in a short time, it is difficult to capture its formation process using experimental methods. Subsequently, the radical can further form chlorite esters with ClO_2_, which are subsequently transformed into o-quinone or o-phthalic acid, p-quinone and muconic acid monoester or their lactones, and release chlorite and hypochlorite [6,7].

However, the complex structure of lignin makes it difficult to study the oxidation process in depth from the molecular perspective. Therefore, the concept of lignin model substances was proposed, and the rational use of lignin model substances for research can effectively solve the difficulties in the study of large molecule lignin, which has become one of the main directions of lignin research, and it helps to further resolve the structural change pattern of lignin from the molecular level [1,8]. Among them, Nie et al. [9] used the lignin model substance 1-(3,4-dimethoxyphenyl)ethanol (MVA). The oxidation reaction of chlorine dioxide on lignin was studied. The results demonstrate that hypochlorous acid is the main cause of AOX formation during bleaching and that the chlorine dioxide reacts with the lignin structure according to level 1 reaction kinetics. Lu et al. [10] studied the structural effects of lignin on its sun protection factor (SPF) UV resistance index, UV absorption and antioxidant .activity using lignin model compounds instead of lignin and found that the presence of conjugated structures and carboxyl groups would significantly affect the UV properties of lignin model compounds. It is thus clear that studies with the help of lignin model compounds are beneficial to fully characterize the structural changes of natural lignin during chemical reactions, with lignin model compounds as a lignin substitute to study its degradation mechanism.

Lignin model compounds [1,8,9] can be classified into different structures [7,11]. The physicochemical properties of lignin are closely related to its structure and functional groups, in particular, the localization effect of different substituents of the benzene ring, methoxy. Methoxy is a key reference indicator of the reactivity of lignin as the main basis for distinguishing lignin structural units, and it has a significant effect on the reactivity of lignin molecules [12,13]. Yang et al. [14] used monosubstituted benzene as a model and the induced effect parameter, conjugation effect parameter, induced effect and conjugation effect were investigated. The results showed that the substituent had the strongest negative effect on the benzene ring when the atom attached to the ring was in the same period as the carbon. When Liang et al. [15] used the conventional mass spectrometry theory, the phenomenon of charge concentration on atoms with lower induced effects was found, and the substitution and induced effects phenomenon was revealed. This is closely related to the electrophilic substitution reaction mechanism. Schultz et al. [16] studied six nonphenolic β-0-4 lignin model species substituted on the benzene ring, and the electron-absorbing substituents that enhance the rate of hydrolysis were revealed, and the results have important implications for the regulation of lignin structure. Hydrogen atoms are susceptible to attack by electrophilic reagents in the presence of π-bonds in benzene rings [17]. This phenomenon leads to the occurrence of electrophilic substitution reactions, especially when a substituent is introduced into the benzene ring, because the electron effect of the substituent is transferred along the benzene ring conjugate chain and the distribution of the electron cloud density on the benzene ring becomes uneven. Therefore, the ease of carrying out electrophilic substitution reactions and the main position of the substituent into the benzene ring will vary with the original substituent [18,19]. The difficulty of electrophilic substitution on the benzene ring and the activation site are changed and the oxidation reaction of lignin with ClO_2_ will be affected. In particular, the uptake process of chlorine dioxide on the hydrogen atoms of phenolic hydroxyl groups is affected, which is an extremely important process in the ClO_2_ bleaching process. Phenol hydroxyl hydrogen atoms are acquired by chlorine dioxide and phenol oxygen radicals are formed in situ. As the initial step of the bleaching reaction, the smooth performance of this reaction is of great importance for the subsequent ClO_2_ oxidation process [20]. However, the study of the mechanism is still unclear.

Therefore, quantum chemical techniques were used in this paper. Six phenolic lignin model compounds with different lignin structures were selected, 4-ethyl guaiacol (4-E-ol), eugenol (E-ol), ferulic acid (FA), 4-hydroxyacetophenone (4-A-ol), 4-hydroxy-3-methoxyacetophenone (APO) and 3,5-dimethoxy-4-hydroxyacetophenone (DHP). By the difference of substituents, the six lignin modelers were classified into two classes, where 4-E-ol, E-ol, FA and APO are G-type lignin modelers with different substituents at the number one position, and 4-A-ol, APO and DHP are lignin modelers with the same substituents at the number one position but with different numbers of methoxy. The process of phenolic oxygen intermediates formed by phenolic lignin structures with ClO_2_ during ClO_2_ bleaching was investigated by density functional theory (DFT) using Gaussian 16. DFT is a method for studying the electronic structure of multielectronic systems. It is one of the most frequently used methods in the field of computational chemistry. The Hammett constants of substituents and the values of dissociation energies of phenolic hydroxyl bonds of benzene rings with different substituents were calculated to investigate the effect of substituents on the reactivity. Electrostatic potential distribution, molecular front orbitals, natural atomic charge distribution, Fukui functions, orbital weight double descriptors and transition state energy barriers were analyzed and the effect of substituents on molecular surface properties during ClO_2_ oxidation of lignin were investigated. The results of this study provide new elements for the efficient oxidative degradation of lignin by chlorine dioxide.

## 2. Results and Discussion

### 2.1. Optimization of Phenolic Lignin Model Compound Structure

The phenolic lignin model species 4-E-ol, E-ol, FA, 4-A-ol, APO, and DHP were selected. In the reaction environment of implicit solvent SMD, the geometric structures of six lignin structures were fully optimized at the m062x/6-311+g (d) level. The structure and substituent information are shown in Table 1. The structural optimization of the energy minimum equilibrium state and the calculation of bond dissociation energy are also performed; they are considered to be important indicators of the strength of chemical bonds [21]. The optimized structure is shown in Figure 1, and the bond dissociation energy data are listed in Table 2. Among the six phenolic lignin model species, 4-E-ol, E-ol, FA, and APO are the ones containing a single methoxy G-type structure, and 4-A-ol and DHP are H- and S-type structures, respectively, which are all widely present in the lignin unit and are the main constituent structures of lignin.

As shown in Figure 1, all six lignin model compounds are composed of C, H, and O elements. The phenolic hydroxyl group is the main reaction site of phenolic lignin model species due to its good nucleophilic activity [8,22]. Through structural optimization, there were differences in the bond lengths of C-O and O-H among the six lignin model compounds. The main manifestations are as follows: the C-O and O-H bond lengths of 4-E-ol are 1.3730 Å and 0.9679 Å, respectively; the C-O and O-H bond lengths of E-ol are 1.3721 Å and 0.9680 Å, respectively; the C-O and O-H bond lengths of FA are 1.3619 Å and 0.9688 Å, respectively; the C-O and O-H bond lengths of 4-A-ol are 1.3607 Å and 0.9668 Å, respectively; the C-O and O-H bond lengths of APO are 1.3591 Å and 0.9690 Å, respectively; and the C-O and O-H bond lengths of DHP are 1.3583 Å and 0.9685 Å, respectively. All six model compounds are Aromaticity molecules. The bond lengths of the C-C bonds are slightly longer than those of the C-H single bonds, and the bond lengths of the C-O single bonds are all between those of the C-H single bonds, which are characteristic of aromatic molecules containing side chains. Especially for the lignin structure containing phenolic hydroxyl group, the phenolic hydroxyl groups react more strongly with electrophilic reagents for two reasons: (1) The good electron absorption characteristic structure of the large π bond in the benzene ring. (2) Localization effect of electron-withdrawing groups (Effect of localized molecular changes on surface properties).

Subsequently, the bond dissociation energies of O-H of the lignin model species were calculated and counted. As shown in Table 2, the bond dissociation energy is an important parameter indicating the strength of a chemical bond, which reflects how strongly two atoms in a molecule are bonded to each other. It is generally believed that the smaller the bond dissociation energy, the worse the stability of the chemical bond, and the easier it is to break [23,24]. The hydroxyl (O-H) dissociation energies of 4-E-ol, E-ol, FA, 4-A-ol, APO, and DHP were 84.38 kJ/mol^−1^, 86.09 kJ/mol^−1^, 87.35 kJ/mol^−1^, 96.73 kJ/mol^−1^, 90.36 kJ/mol^−1^, and 86.19 kJ/mol^−1^, respectively. It is clear from the results in Table 3 that the Hammett constants calculated using the NBO charge were 2.91, 2.86, 1.32, −0.17, 0.27, and 0.70, respectively. The results of the Hammett constant indicate that the substituents influence the reactivity of the lignin structure, showing a correlation with the dissociation energy of the hydroxyl group. The larger the Hammett constant is, the smaller the relative energy of O-H bond dissociation in the hydroxyl group. This indicates that the difference in electron-absorbing and electron-donating properties of the substituents influences the initial degradation reaction of the lignin structure and elaborates that the selection of the six lignin model species is representative.

Different amounts of methoxy influence the hydroxyl bond dissociation energy when the substituents are the same. H-type lignin has the highest hydroxyl dissociation energy, followed by G-type and finally S-type. This indicates that the uniform electron distribution improves the stability of the molecule when no methoxy is present in the benzene ring, making the bond dissociation difficult. On the contrary, when the number of Methoxy group groups increases, the uniform electron cloud distribution characteristics are destroyed. Stable and electron withdrawing Methoxy group groups exist on both sides of the hydroxyl group, which increases the potential difference of the hydroxyl group. Moreover, hydroxyl groups serve as ortho para substituents, enhancing the activity below the benzene ring [25]. The results illustrate that the ether bonds attached to G- type and S-type lignin are more prone to breaks in the lignin structure than in the H-type lignin structure. This stems from the molecular polarity difference, which can be accounted for by the difference in the bond dissociation energy data.

When the substituents are different, the structure of the substituent at position 1 has an important effect on the hydroxyl group dissociation energy, which may be due to the difference in the potential difference between the oxygen and hydrogen atoms in the hydroxyl group of the substituent [12,13]. When position 1 is an electron-rich ethyl structure, the potential difference of the hydroxyl group is large. With the increase of electron-absorbing groups at position 1 (oxygen atom and C-C double bond), it causes the decrease of the potential difference of the hydroxyl group, which leads to the increase of the bond dissociation energy. Chain length and bond dissociation energy also show a negative correlation. The longer the chain length, the lower the bond dissociation energy.

### 2.2. Effect of Different Benzene Ring Branched Substituents on the Uptake of Lignin Hydroxyl Hydrogen Atoms by ClO_2_

#### 2.2.1. Structural Optimization of Phenolic Lignin-ClO_2_ Transition State with Different Benzene Ring Branched Substituents

The transition state structures formed by lignin modelers with different benzene ring branched substituents with ClO_2_ are listed in Figure 2 for 4-E-ol, E-ol, FA and APO. The six lignin structures were fully optimized at m062x/6-311+g (d) level in the reaction environment of implicit solvent SMD. After frequency analysis, all compounds are present and only one imaginary frequency, so the obtained transition state structures are accurate and reliable. The main bond length and bond angle data are listed in Table 4.

From the results, it can be seen that the uptake of hydrogen ions in phenolic hydroxyl groups by ClO_2_ will cause changes in bond length. As a strong free radical oxidant with electrophilic properties, ClO_2_ has a great electron absorption effect [26]. The C-O bonds of the four model species grew to 1.6243 Å, 1.6171 Å, 1.5691 Å and 1.6978 Å, respectively. The bond lengths of O-H bonds grew to 1.0743 Å, 1.0096 Å, 1.0148 Å and 1.0296 Å, respectively. This represents the gradual attraction of the hydrogen ion away from the benzene ring through the uptake process of the electrophilic reagent. ClO_2_ formed bond strengths similar to the C-C bond with hydrogen ions with bond lengths of 1.0429 Å, 1.0096 Å, 1.0148 Å and 0.9983 Å, respectively. The formation of the transition state is also corroborated by the bond length variation.

#### 2.2.2. Electrostatic Potential Distribution of Phenolic Lignin-ClO_2_ Transition State with Different Benzene Ring Branched Substituents

The molecular surface electrostatic potential (ESP) is an important research indicator in quantum chemistry studies. It is often used to determine the specific properties of a molecule. The electrostatic potential is related to the reactivity of the molecule, so the distribution of the electrostatic potential around the molecule varies with the space around the molecule [27]. The electrostatic potential represents the work done to move a unit positron from infinity to somewhere in the space around a molecule, and the magnitude of this indicator is of great importance in understanding intermolecular forces. It is known from current research that during chemical molecular reactions, non-covalent interactions between molecules contribute to the occurrence of chemical reactions [28], the electrostatic interaction is the main mechanism, so the molecular surface electrostatic potential is particularly important for the chemical reaction process. To improve the quantitative accuracy, the difference in electrostatic potential projected onto the molecular surface by different colors is usually used as a basis for examining the distribution of electrostatic potential on the molecular surface. The most likely site for electrophilic (nucleophilic) reactions is the location of the electrostatic potential minimum (maximum point) on the molecular surface, which can be analyzed using quantitative molecular surface analysis algorithms. In this experiment, four phenolic lignin structures were molecularly optimized, and the data were analyzed using Multiwfn 3.8 wave function analysis software. Combined with VMD, the molecular surface electrostatic potential distribution map was plotted, In addition, the quantitative distribution of the molecular surface electrostatic potential of the three phenolic lignin structural monomer molecules was plotted, as shown in Figure 3.

As can be seen from Figure 3, the benzene ring and the oxygen atom are the main electron-absorbing groups [29], which are distributed in the blue region. Alkyl, as the main electron donating group, is distributed in the red region. At the phenolic hydroxyl group, there is a certain amount of potential difference, which makes it easy to be attacked by electrophilic reagents. The quantitative information on the distribution of the molecular electrostatic potential reveals that when the electron-absorbing group of the first substituent increases, the electron-donating region of the molecule increases and has good electron-donating properties and is more prone to electrophilic oxidation reactions. This represents that the lack of electron substituents affects the distribution of the molecular electrostatic potential. This is consistent with the trend of the dissociation energy of phenolic hydroxyl bonds.

Subsequently, the reaction of ClO_2_ with phenolic lignin model species reveals the formation of transition states and the trapping of hydrogen ions by ClO_2_. When the transition state is formed, the electron-absorbing property of ClO_2_ directly affects the equilibrium of the molecular electron cloud, causing the electron-deficient region to flow toward the transition state formation region [30]. Combined with Figure 3 it is clear that the area of the region with positive electrostatic potential increases. It indicates that activation of the lignin molecule may have occurred under the influence of ClO_2_, making the molecule highly susceptible to attack by excess electrophilic compounds and electrophilic substitution reactions. By comparing the structures of phenolic lignin with four different substituents at position 1, it can be found that the substituents with shorter branches and strong electron-absorbing ability are more stable and less susceptible to the influence of ClO_2_. And the flow of electron clouds is more pronounced for molecules with longer branched chains. This indicates that ClO_2_ can influence the electron arrangement around the molecule, which directly affects the electrophilic activity of the molecule. Especially under high concentration of the ClO_2_ reaction system, it provides a possible reaction site for the free ClO_2_.

#### 2.2.3. Natural Atomic Charges of Phenolic Lignin-ClO_2_ Transition States with Different Benzene Ring Branched Substituents

An atom is a charged system with a charged structure. The point charge at the center of the atom is usually taken as the natural atomic charge of the atom, which is one of the simplest, most intuitive, and fundamental ways to describe the charge distribution in a chemical system. It is commonly used for predicting reaction sites in chemical reaction processes, and corresponding prediction models can be established through charge analysis to provide a reference for researchers. It has been suggested that the Hirshfeld charge calculation method is more suitable for most chemical reactions and represents the summation of the electron density of all atoms in the free state [31]. The calculation of atomic charges can be useful for quantifying the structure and properties of molecules, it is of great significance to analyze the reaction of molecules and atoms in the process of chemical reaction. In this study, natural atomic charges were calculated at the m062x/6-311+g (d) level for four phenolic lignin model species and their transition states.

The hydrogen atoms of all four lignin modeler molecules are positively charged and most of the carbon atoms are negatively charged, due to the major benzene ring backbone. This is because the electronegativity of the C atom is greater than that of the hydrogen atom, making the C atom more capable of absorbing electrons. However, the electronegativity of the oxygen atom is greater than that of the carbon atom, so that the negative current of the carbon atom connected to the oxygen atom flows to the oxygen atom, which has a greater negative charge, resulting in a negative charge of the carbon atom connected to the oxygen atom. This is consistent with the analysis of the electrostatic potential above. The large π-bond strengthens the electron-absorbing effect of the molecule, and the electron-pushing effect of the hydroxyl group strengthens the polarity of the O-H bond, so that the hydrogen of the hydroxyl group in phenol can be ionized [32]. Statistical analysis of the charge differences of phenolic hydroxyl groups in four different lignin model compounds with substituents at the position 1, The calculated charge differences are 0.965, 0.963, 0.944, and 0.942, respectively. The charge difference gradually decreases, which indicates that the hydrogen ions are progressively less active and less accessible to electrophilic reagents, which is consistent with the results of the bond dissociation energy. When chlorine dioxide takes in hydroxyl hydrogen atoms, the atomic charge of the oxygen atom changes, and the trend of charge change is from −0.521 to −0.535, −0.520 to −0.572, −0.494 to −0.525, −0.491 to −0.516, respectively. The results demonstrate that the number of negative charges increases after the uptake of hydrogen atoms by chlorine dioxide. This not only proves the transition state formation, but also states the concentration of negative charges to this region during the formation of phenoxy radicals. Finally, by comparing the natural atomic charges of the molecules, we derived the ranking order of this reactivity as APO < FA < E-ol < 4-E-ol, respectively. This is in accordance with the previous results of bond length comparison and electrostatic potential analysis.

#### 2.2.4. Characterization of the Front-Line Orbital Distribution of Phenolic Lignin-ClO_2_ Transition States with Different Benzene Ring Branched Substituents

Frontline orbital theory as a basic molecular orbital theory dictates that the electron cloud distributed around a molecule is subdivided into different orbitals depending on the energy difference. The HOMO orbitals represent the highest-energy molecular orbitals and those not occupied by electrons, and the LUMO orbitals represent the lowest-energy molecular orbitals [33]. Therefore, the chemical reactions occurring in a system are mainly determined through LUMO orbitals. So the frontline orbital theory has excellent performance for the activity of redox reaction systems [34]. In this experiment, four phenolic lignin model objects were used as research objects, and HOMO-LUMO frontier orbitals of four phenolic lignin and ClO_2_ transition state structures were drawn, as shown in Figure 4. The specific frontier orbital energy information was listed in Table 5.

Figure 4 shows that the HOMO orbitals of the phenolic lignin monomer model species are mainly distributed on the oxygen atoms of the adjacent hydroxyl groups. LUMO orbitals are easily accessible to electrons and are mainly distributed on carbon and oxygen atoms. HOMO orbitals are prone to lose electrons. Since the electrons in the orbitals are easily excited and can easily escape from the atomic bindings, electrophilic reagents are highly susceptible to such atoms. LUMO orbitals are easily accessible to electrons and susceptible to attack by nucleophilic reagents. Therefore, the phenolic hydroxyl group of the phenolic lignin structure is easily attacked by electrophilic reagents, resulting in electrophilic removal reactions, and H atoms are more easily attracted by electrophilic reagents. In addition, the LUMO orbitals are mainly distributed in the aromatic ring formation, so the electron flow is affected to the extent that the bond between the carbon atom and the hydroxyl group is weakened. Therefore, the structure has a higher activity in the presence of electrophilic reagents. The positions of the easily accessible electrons and the electron aggregation can be seen in the figure. It can be observed that the HUMO orbitals of the molecule are concentrated around the carbon atom on the benzene ring and the oxygen atom attached to the benzene ring. And the LUMO orbitals of this molecule are mainly concentrated near position 1 where the benzene ring of this molecule is attached to the branched chain. This is slightly different from the orbital distribution of APO. So, it indicates that the molecule loses electrons mainly in the phenolic hydroxyl region, and gains electrons mainly in the side chain at position 1. It also revealed that the molecule is highly reactive to the attack of electrophilic reagents.

#### 2.2.5. Distribution Characteristics of Fukui Functions and Double Descriptors for Phenolic Lignin-ClO_2_ Transition States with Different Benzene Ring Branched Substituents

For the prediction studies of reactive sites, the electrostatic potential mainly describes the effect of mutual attraction between molecules through electrostatic interactions. It is worth noting that due to the local nature of chemical reactions, the activity of molecules at different positions varies. Therefore, in the process of predicting reaction behavior, it is important to be able to identify and rank the electrons that have the least firm hold, the most available electrons, and the positions where those electrons are tightly bound. This requires not focusing on specific electron orbitals, which are usually somewhat delocalized rather than focusing on specific points in molecular space, although electrons from several different orbitals may have a high probability of being at each such point. To allow a quantitative and accurate analysis of the electrophilic or nucleophilic activity of the system, the concept of local softness was proposed, which allows a more accurate evaluation of the electrophilic activity of the molecule. This experiment interprets the local reactivity of organic molecules by contracting the Fukui function to each atom of a compound through the concept of condensed Fukui function (CFF) [35].

Based on the wave function analysis software Multiwfn 3.8, the extreme value points of the reduced Fukui function and orbital weight double descriptors were calculated for three different methoxy base lignin and its Transition state. Both the reduced Fukui function and the double descriptor are analyzed using the Hirshfeld charge values as a guide. The reasonableness of the Hirshfeld charge for such calculations has been tested [36]. The equation is calculated as follows, setting the corresponding electron density of the equivalent surface to 0.01 au. For ease of observation, the very small value points and a small number of very large value points values are marked. The values of f-, f+, and f0 for each atom in a molecular compound can be accurately calculated through formulas. Also, the main double descriptor minima points are listed.
(1)$$\mathrm{Electrophilic}\;\mathrm{reaction}:f_{A}^{-}=q_{N-1}^{A}-q_{N}^{A}$$
(2)$$\mathrm{Nucleophilic}\;\mathrm{reactions}:\;f_{A}^{+}=q_{N}^{A}-q_{N-1}^{A}$$
(3)$$\mathrm{Free}\;\mathrm{radical}\;\mathrm{reaction}:\;f_{A}^{-}=(q_{N-1}^{A}-q_{N+1}^{A})/2$$

According to the calculation results, it can be seen that the four phenolic lignin structures with different substituents at the first position have high nucleophilic activity at the phenolic hydroxyl group, which is easily attacked by electrophilic reagents [36], The charge on the benzene ring was changed after the uptake of the hydrogen ion of the phenolic hydroxyl group. C1–C6 and C3–C4 are the active sites of electrophilic reactions before the formation of oxygen radicals. However, the electrophilic active site shifts to the neighboring counterpart of the oxygen radical when the oxygen radical is formed. Apparently, the electrophilic activity at the C3 position diverged to the neighboring position. The electrophilic activity of the parasite was aggregated. The probability of benzoquinone formation was increased. The three molecules exhibited the same reaction characteristics. In addition, the electrophilic activity of the C1 position was more pronounced when the substituent contained an oxygen atom and a double bond-like electron-absorbing structure. This means that the branched chains are preferentially oxidized to form oxygen-containing double-bond structures. The conversion of lignin to intermediates is facilitated during the oxidation of lignin by chlorine dioxide. The f-values of the C atoms attached to the phenolic hydroxyl groups are 0.1235, 0.1211, 0.1087, and 0.1045, respectively. The f+ values of C atoms are 0.0771, 0.0735, 0.0469, and 0.0353, respectively. The f-values of O atoms are 0.1178, 0.1155, 0.1071, and 0.1059, respectively. The f+ values of O atoms are 0.0234, 0.0298, 0.0203, and 0.0189, respectively. The magnitude of the value exhibits a positive correlation with the probability of gaining and losing electrons. The comparison reveals that the ease of occurrence of the electrophilic reaction is that APO < FA < E-ol < 4-E-ol. This is like the results of the previous study. It is suggested that the electron-absorbing groups of lignin-branched chains play a passivating role in the lignin molecule.

The orbital weight double descriptor is analogous to the mechanism studied for the Fukui function [37]. It serves as an important method for the quantitative detection of molecular surface properties and can effectively predict chemical reaction active sites. Based on previous studies, it has been shown that orbital-weighted double descriptors are reliable for the prediction of electrophilic and nucleophilic reaction sites [38]. Figure 5 shows the extreme-point data information of the track-weight double descriptor for four different substituents and their transition states.

The local softness allows a more accurate evaluation of the electrophilic activity of the molecule [39]. The Hirshfeld charge minima described by the double descriptor are basically distributed around the carbon and oxygen atoms. Among them, there are maximum points at both ends of ClO_2_′s oxygen atom. It is electrophilic and easily encounters electron-rich groups, which can quickly form an electron equilibrium to the lowest energy state. This is the reason for the rapid reaction of ClO_2_ with phenolic structures. The hydroxyl group is an ortho-para-substituent. In this reaction system, the adjacent and para-carbon atoms of the hydroxyl group obviously have corresponding minimum points. This indicates that the electron binding force near the adjacent and para-carbon atoms is the weakest, which is the most likely reactive site. Among them, the activity ranking is C6 > C2 > C4. The presence of minima near O10 indicates an increased chance of electrophilic reactions around the oxygen atom due to the influence of lone pairs of electrons that receive easy polarization. However, it is less active and significantly larger than the other minimal value points. This also confirms that the electrophilic reaction should take place on the hydroxyl hydrogen atom and not on the oxygen atom.

Along with the uptake of hydroxyl hydrogen atoms by ClO_2_, the minimal value points of the hydroxyl neighboring, and opposite positions increase, indicating an increase in the electrophilic activity of this position. From the numerical comparison, we derived the ranking order of this reactivity as APO < FA < E-ol < 4-E-ol. This is the same result as the previous analysis, indicating that the influencing factor of molecular activity is mainly the distribution pattern of electrons, and the localization effect of substituents is related to the potential difference.

#### 2.2.6. Calculation of Energy Barriers for Phenolic Lignin-ClO_2_ Transition States with Different Benzene Ring Branched Substituents

By analyzing the differences in the molecular structures of the four lignin model substances, it is clear that the differences in structure inevitably affect the energy differences in the formation of the transition state. For example, molecules with low energy tend to have better symmetry. In this experiment, the energy barriers for the formation of transition states of phenolic lignin molecules with four different substituents at the number one position were mainly calculated and compared. The specific values are listed in Table 5.

As can be seen from Table 6, the four lignin structure molecules require lower energy and react faster in the formation of intermediates. All of them are extremely easy to make react with ClO_2_ and participate in the formation process of intermediates. Among them, the energy barriers of 4-E-ol and E-ol molecules are close and low, with values of 19.5153 kcal/mol and 19.1224 kcal/mol, respectively. This implies that the molecule is more likely to react than the other two molecular structures. This was followed by FA with a value of 20.6040 kcal/mol. Finally, the reaction energy barrier of the APO molecule is 21.4906 kcal/mol. This agrees with the analysis presented in the previous section, which clarifies the role of substituents on the lignin oxidation reaction activity.

### 2.3. Effect of Methoxy on the Uptake of Lignin Hydroxyl Hydrogen Atoms by ClO_2_

#### 2.3.1. Structural Optimization of Phenolic Lignin-ClO_2_ Transition States with Different Methoxy Numbers

Three phenolic lignin model compounds of guaiacyl type (G), syringyl type (S), and p-hydroxyphenyl type (H) structures were selected as 4-hydroxy acetophenone, 4-hydroxy-3-methoxyacetophenone and 3,5-dimethoxy-4-hydroxyacetophenone, respectively. The effect of the amount of methoxy and its substitution position on the molecular activity and the oxidation of ClO_2_ was investigated. According to the characteristic structure of lignin, G, S, and H type is the basic configuration of lignin. The high molecular weight lignin macromolecules consists of three structural units that are tightly bound together by aryl ether bonds. Among them, the lignin in hardwood biomass mainly contains G-types and S-types. Lignin in coniferous wood biomass is mainly composed of G-type. Lignin in graminaceous plant biomass is composed of G-types, S-types, and H-types. Therefore, the study of the reactivity of lignin containing different methoxy is very important, especially in the study of oxidative degradation of lignin, which provides a theoretical basis for the high-value utilization of lignin.

Therefore, in this experiment, 4-A-ol, APO, and DHP were selected as the research objects, and the geometric structure of six lignin structures was fully optimized at m062x/6-311+g (d) level in the reaction environment of implicit solvent SMD. After frequency analysis, all compounds are present and only one imaginary frequency is present, so the obtained transition state structures are accurate and reliable. The transition state structures formed by phenolic lignin modelers with different numbers of methoxy with ClO_2_ are listed in Figure 6, and the main bond length and bond angle data are listed in Table 7.

The product structures of the transition states formed by the lignin modelers with different methoxyl groups with ClO_2_ are listed in Figure 6, and they are 4-A-ol, APO, and DHP. According to the structural optimization results, the uptake of hydrogen ions in phenolic hydroxyl groups by ClO_2_ will cause changes in bond length. ClO_2_, a strong radical oxidizer with electrophilic nature, has great electron absorption and the carbon-oxygen bonds of the four model species grow to 1.6352 Å, 1.6978 Å, and 1.2499 Å, respectively. Meanwhile, the bond lengths of O-H bonds grew to 1.0041 Å, 1.0296 Å, and 1.4509 Å, respectively. This represents that hydrogen ions gradually move away from the attraction of the benzene ring through the uptake process of electrophilic reagents. ClO_2_ formed bond strengths similar to C-C bonds with hydrogen ions, and their bond lengths were 1.0041 Å, 0.9983 Å, and 1.0404 Å. The formation of the transition state is also corroborated by the bond length variation.

#### 2.3.2. Characteristics of Electrostatic Potential Distribution of Phenolic Lignin-ClO_2_ Transition States with Different Numbers of Methoxy

The molecular optimization of the three phenolic lignin structures and their transition states was carried out and the molecular surface electrostatic potential distributions were analyzed and plotted using Multiwfn combined with VMD. The quantitative distribution of the molecular surface electrostatic potential of the three phenolic lignin structure monomer molecules was plotted, as shown in Figure 7.

As can be seen from the figure, the p-hydroxyphenyl type lignin has a uniform electrostatic potential distribution due to the absence of methoxy in the neighboring position of the phenolic hydroxyl group, and the distribution of positive and negative electrostatic potential is similar. When a methoxy is added to the neighboring position of the phenolic hydroxyl group, the area of the negative electrostatic potential region decreases substantially and the positive electrostatic potential region is concentrated on the methoxy side, which leads to the appearance of a large potential difference. When methoxy is added simultaneously at two neighboring positions of the phenolic hydroxyl group, it leads to an increase in the area of the negative region of the electrostatic potential due to the increase in oxygen atoms compared to the G-type lignin structure. Correspondingly, the values in the regions with more positive electrostatic potential are reduced, but still dominate the area region, indicating that the molecule still has a strong electron supply capacity.

Subsequently, the reaction of ClO_2_ with phenolic lignin model species reveals the formation of transition states and the trapping of hydrogen ions by ClO_2_. When the transition state is formed, the electron-absorbing properties of ClO_2_ directly affect the equilibrium of the molecular electron cloud, causing the electron-deficient region to flow to the region formed by the transition state. Combined with Figure 7, it is clearly found that the area of the region with positive electrostatic potential increases. Similar to the previous analysis, activation of the lignin molecule may have occurred under the influence of ClO_2_, making the molecule highly susceptible to attack by excess electrophilic compounds and electrophilic substitution.

Unlike the substituent at position 1, methoxy plays a role in the lignin structure to stabilize the structure and increase the activity of the phenolic hydroxyl group. The electron-absorbing property of the oxygen atom in methoxy increases the potential difference between the oxygen and hydrogen atoms in the phenolic hydroxyl group, resulting in the ionization of H ions. Combined with the previous data on bond dissociation energy, it can be seen that the number of neighboring methoxy is inversely proportional to the magnitude of the dissociation energy. In addition, the methoxy group also provides directivity for the oxidation reaction. Looking at the electrostatic potential distribution of the reaction process of the three molecules with ClO_2_, 4-A-ol is more likely to undergo substitution at the para-quinone position and form a para-quinone. The G-type lignin structure has an increased negative region of electrostatic potential, and the probability of degradation reaction is lower than that of the S-type structure after the uptake of H atoms. This could explain the greater difficulty of G-type lignin to be degraded compared to S-type lignin, stemming from the difference in the distribution of electrostatic potential.

In general, methoxy can effectively promote the activity of hydroxyl groups, increasing the rate of electrophilic reactions. Methoxy has a certain localization effect, the benzene ring of H-type lignin and G-type lignin is not easily destroyed, and the chlorination reaction probability is low, and the degradation is mainly through the oxidation process.

#### 2.3.3. Natural Atomic Charges of Phenolic Lignin-ClO_2_ Transition States with Different Methoxy Numbers

Natural atomic charges were calculated at the m062x/6-311+g (d) level for the three phenolic lignin model species G, S, and H and their transition states.

The hydrogen atoms of all three lignin modeler molecules are positively charged and most of the carbon atoms are negatively charged due to the major benzene ring backbone. This is because the electronegativity of the C atom is greater than that of the hydrogen atom, making the C atom more capable of absorbing electrons. However, the electronegativity of the oxygen atom is greater than that of the carbon atom, so the negative current of the carbon atom connected to the oxygen atom flows toward the more negatively charged oxygen atom, resulting in a negative charge of the carbon atom connected to the oxygen atom. This is consistent with the analysis of the electrostatic potential above. The phenolic hydroxyl charge differences of the three lignin model substances with different methoxy contents were calculated to be 0.907, 0.942, and 0.956. The charge difference gradually increases, which indicates that the hydrogen ions are progressively more reactive and less accessible to electrophilic reagents, which is consistent with the analysis of the bond dissociation energy. By comparing the natural atomic charges of the molecules and starting only from the analysis of the reactant structures themselves, we arrived at this ranking order of reactivity, 4-A-ol < APO < DHP. This is in line with both the previous bond length comparison results and the electrostatic potential analysis results. The simultaneous activation of the benzene ring by methoxy and the reaction localization effect, is an essential reaction mechanism in the oxidation of ClO_2_.

#### 2.3.4. Characterization of the Front-Line Orbital Distribution of Phenolic Lignin-ClO_2_ Transition States with Different Methoxy Numbers

In this experiment, the HOMO-LUMO front orbitals of three phenolic lignin model species with different methoxy quantities were studied using Multiwfn to plot the HOMO-LUMO front orbitals of the three phenolic lignin and ClO_2_ transition state structures were plotted as shown in Figure 8, and the specific front orbitals energy information is listed in Table 8.

The distribution of the highest occupied orbitals (HOMO) and the lowest air channel (LUMO) for the three phenolic lignins and their transition states are presented in Figure 8. The results of wave function analysis reveal that the HOMO orbitals of the three phenolic lignin molecules are mainly concentrated on the benzene ring and oxygen atom, which belong to the electrophilic effect of electron-absorbing groups. The LUMO orbitals are mainly clustered on the inter-parallel carbon atoms of the phenolic hydroxyl group, indicating that the phenolic hydroxyl group is an active site for electrophilic reactions. Combined with the detailed data in Table 7, the HOMO-LUMO gaps are 7.2413 eV, 6.8342 eV, and 6.7321 eV. This suggests that methoxy is beneficial for improving molecular stability when the basic structure is the same.

When ClO_2_ molecules are introduced, the HOMO orbitals of H-type and G-type lignin tend to concentrate towards the carbon atom at position 1, indicating that electrophilic reagents may be guided by the concentration of HOMO orbitals. However, the orbital distribution of the S-type lignin structure did not change significantly, which laterally illustrates the stabilizing effect of methoxy. The change in gap also demonstrates that the G type is more elevated, followed by the H type, and the S type is decreasing. This indicates that H-type and G-type are the main activated degradation groups under the oxidation of ClO_2_. H-type tends to the formation of quinone, and G-type is more prone to the reaction of demethoxylation.

#### 2.3.5. Characteristics of Fukui Functions and Double Descriptor Distribution of Phenolic Lignin-ClO_2_ Transition States with Different Numbers of Methoxy

Based on the wave function analysis software Multiwfn 3.8, the reduced Fukui function and orbital weight double descriptor extreme points were calculated for three different methoxy numbers of lignin and their transition states. Both the abbreviated Fukui function and the double descriptor are analyzed with the Hirshfeld charge value as a guide, and the formula is given in Section 2.2.5, setting the equivalent face corresponding to an electron density of 0.01 au. For ease of observation, the very small value points and a few very large value points are marked. The values of f-, f+, and f0 for each atom in a molecular compound can be accurately calculated by the formula. Also, the main double descriptor minima are listed in Figure 9.

According to the calculations, it is clear that all three phenolic lignin structures with different methoxy numbers have high nucleophilic activity at the phenolic hydroxyl group and are easily attacked by electrophilic reagents. It can be seen that the f-values of the C atoms attached to the phenolic hydroxyl groups are 0.1173, 0.1045, and 0.1310, respectively. The f+ values for C atoms are 0.0714, 0.0701, and 0.0687, respectively. The f-values of O atoms are 0.1451, 0.1059, and 0.1359, respectively. The f+ values of O atoms are 0.0327, 0.0330, and 0.0333, respectively. The magnitude of the values showed a positive correlation with the probability of gaining and losing electrons, and the comparison showed that the ease of occurrence of the electrophilic reaction was, 4-A-ol < APO < DHP. Similar results to the previous analysis. The Hirshfeld charge minima described by the double descriptor are basically distributed around the carbon and oxygen atoms. Among them, the presence of extreme value points at both ends of the oxygen atom of ClO_2_ is electrophilic and easily comes into contact with electron-rich groups. This signifies that the electron equilibrium process is accelerated to the lowest energy state, which is the reason for the rapid reaction of ClO_2_ with phenolic structures. In the absence of methoxy present, the phenolic hydroxyl group embodies the basic properties of the o-para substituent. When the lignin type is G, the double descriptor minima are shifted in the methoxy direction. When the benzene ring has a dimethoxy on it, the electrophilic active site moves to the interposition. A clear double descriptor extreme value point appeared at the methoxy site, and the activity of the electrophilic substitution was reduced, effectively indicating a trend toward stabilization of the benzene ring. When the ClO_2_ molecule was introduced, the distribution of the pole points appeared to be significantly changed. Firstly, the minimum points of the H-type structure are mainly distributed at the first position of the benzene ring, indicating that when ClO_2_ was saturated, the free ClO_2_ molecules mainly attacked H-type lignin in the para position and underwent oxidative degradation through electrophilic substitution. The Methoxy group and the substituent at position 1 of G-type lignin have electrophilic activity, and the values are 0.0014450 and 0.0010345 respectively. The values are relatively close, indicating that the response probabilities are comparable. The S-type lignin showed an adjacent para-position with three active sites characterized by values of −0.0010487, −0.0006461, and −0.0009961 for C2, C4, and C6, respectively. It indicates that the substitution probability at position 1 is lower compared to demethoxylation, but the values are all smaller than the G-type structure, which suggests that the G-type is the main degradation unit in lignin degradation and the S-type is more stable.

#### 2.3.6. Calculation of Energy Barriers for Phenolic Lignin-ClO_2_ Transition States with Different Numbers of Methoxy

By analyzing the differences in the molecular structures of the four lignin model species, it is clear that the differences in structure will inevitably affect the differences in energy during transition state formation. For example, molecules with low energy tend to have better symmetry. In this paragraph, the energy barriers for the formation of transition states of three phenolic lignin molecules with different numbers of methoxy were calculated and compared. The specific values are listed in Table 9.

As can be seen from Table 9, the three lignin structural molecules, with the same conclusion as the previous calculations, require lower energy for the intermediate product formation process and the reaction occurs faster, and all are extremely susceptible to the intermediate product formation process by reaction with ClO_2_. The difference in energy barriers is consistent with the trend of phenol hydroxyl dissociation energy. The highest energy barrier for the formation of a transition state is 23.6598 kcal/mol for 4-A-ol, followed by APO with 21.4906 kcal/mol. The lowest energy barrier is 19.2268 kcal/mol for the transition state of the S-type structure.

## 3. Methods and Materials

All quantum chemical calculations involved in this experiment were performed by the Guassian 16 computational package [40]. The geometric full optimization of the six lignin structures was performed at the m062x/6-311+g (d) level with the help of the density generalized function theory (DFT) system to obtain stable structures of the monomers. The m062x method has been widely studied as a common calculation method in recent years. It has been widely used in the study process of common organic systems as well as electrophilic oxidation of organic structures [41,42,43]. Moreover, as a quantum chemistry theory, DFT is widely used, which has the advantages of fast calculation speed and high calculation accuracy [44,45]. The calculation conditions of this experiment were 298.15 K and 101.33 Pa. With the help of transition state theory, the transition state search combined with the structural optimization calculation method (opt +TS) was used to predict the transition state structure at the beginning of the study. Frequency analysis was conducted at the same level. High-precision energy calculations were performed on monomers and intermediate products at the m062x/aug-cc-pVQZ level to obtain the transition structure energy barrier of free radical intermediate products [42,46]. The obtained structure has only one imaginary frequency through vibration frequency analysis. After the stable monomer structure was obtained, the Multiwfn 3.8 (A quantum chemical wave function analysis program) was used to perform the wave function analysis of the molecular structure [47]. And visualization of frontline molecular orbitals and molecular electrostatic potential (ESP) surfaces was accomplished using the Visual Molecular Dynamics program (VMD) [48]. Additionally, Molecular surface electrostatic potential [49], frontline molecular orbit [50], Natural atomic Hirshfeld charge [31], Fukui Function, orbital weighting double descriptor, and transition state energy barrier were analyzed. The Hammett constants of phenolic lignin structures with different substituents were obtained from natural atomic charge calculations. The substituent Hammett constant can be obtained by substituting the natural atomic charge into the linear relation (α = 3.762 + 18.469q R^2^ = 0.94) between the charge on the para-carbon atom of the benzene ring modified by conventional organic substituents. To ensure the stability of the reaction environment, all reactions were carried out in an implicit solvent (SMD) environment [51].The SMD solvent model can represent the average effect of solvents and guarantee low computational time-consumption. Therefore it is widely used in quantum chemistry and molecular simulation. The bond dissociation energy discussed in this study was the enthalpy change between the radicals formed by the homogeneous cleavage of chemical bonds and the reactants. The bond dissociation energy is calculated as:*H = ε_ele_ + ZPE + ΔH_0→298.15k_*(4)
*E_0_(A-B) = H(A^·^) + H(B^·^) − H(AB)*(5)

In the equation, *H* represents enthalpy, *ε_ele_* is the ground state energy of the electron, *ZPE* is the zero-vibration energy, and *E*_0_ represents the bond dissociation energy when the zero-vibration energy is corrected.

## 4. Conclusions

The molecular properties of six lignin model species reacting with ClO_2_ to form transition states were compared by applying the m062x method of DFT at the 6-311+g (d) basis group level. The structural features and reaction characteristics of ClO_2_ oxidation of phenolic lignin model compounds with different numbers of substituents and methoxy groups at position 1 were systematically investigated. The differences in electrostatic potential, atomic charge, Fukui function, orbital weight double descriptor, and reaction energy barrier were compared. The results revealed that activation of lignin molecules occurred under the influence of ClO_2_, making the molecules highly susceptible to attack by electrophilic compounds free in the system and increasing the probability of electrophilic substitution reactions. By comparing the structures of phenolic lignin with four different substituents at position 1, it can be observed that the substituents with shorter branched chains and strong electron absorption are more stable and less susceptible to the influence of ClO_2_, and the flow of electron clouds is more pronounced for the molecules with longer branched chains. This demonstrates that ClO_2_ can influence the electron arrangement around the molecule, which directly affects the electrophilic activity of the molecule. Especially under a high concentration of ClO_2_ reaction system, it provides a possible reaction site for the free ClO_2_. The electron-withdrawing effect of the methoxy group results in a low dissociation energy of the phenol hydroxyl group, making it easier for electrophiles to attack this reaction site. The combined results concluded that methoxy also played a role in stabilizing the molecular structure of the lignin monomer. In general, the substituents have a large effect on the degradation of lignin. G-type lignin is more likely to form chlorite structures and thus be degraded. The phenomenon of para-activation of H-type lignin structure resulted in the tendency of quinone oxidation product formation being increased. In the lignin macromolecular structure with interlinked phenyl ether bonds, the ether bonds linked to the S-type structure are more easily broken. And due to the presence of double methoxy, the lignin structural unit is more stable and less prone to ring opening.

## Figures and Tables

**Figure 1 ijms-24-11809-f001:**
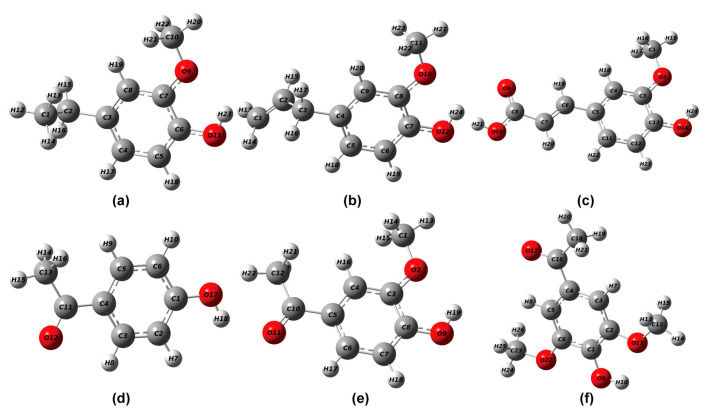
Geometric optimization configuration of lignin phenolic model (**a**) 4-ethylguaiacol; (**b**) eugenol; (**c**) ferulic acid; (**d**) 4-hydroxyacetophenone; (**e**) 4-hydroxy 3-methoxyacetophenone; (**f**) 3,5-dimethoxy-4-hydroxyacetophenone.

**Figure 2 ijms-24-11809-f002:**
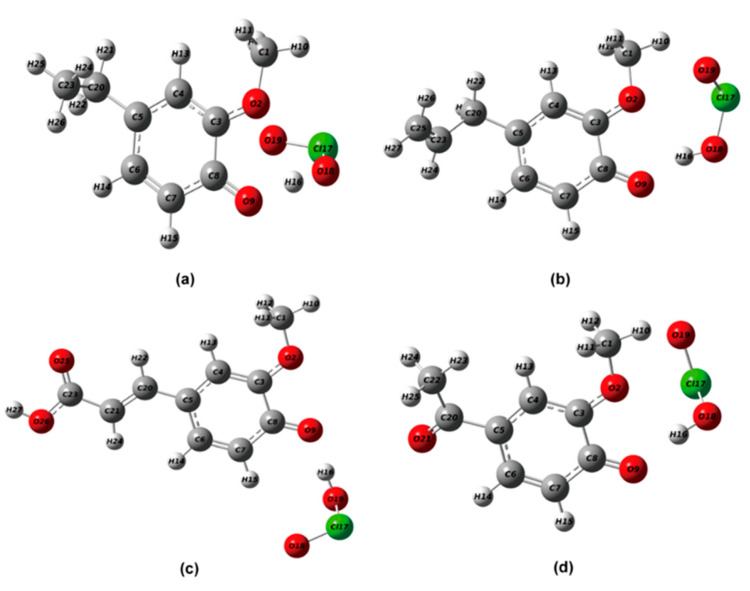
Geometrically optimized transition state structure of phenolic lignin model substance (four different substituents at position 1) with chlorine dioxide (**a**) 4-E-ol; (**b**) E-ol; (**c**) FA; (**d**) APO.

**Figure 3 ijms-24-11809-f003:**
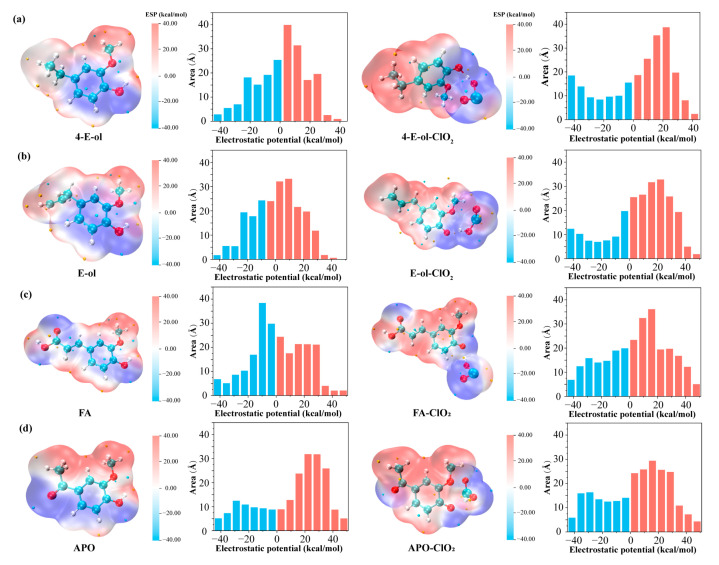
Quantitative distribution of electrostatic potential of lignin phenol model compounds with different substituents at position 1 and lignin chlorine dioxide transition state structure, (**a**) 4-E-ol; (**b**) E-ol; (**c**) FA; (**d**) APO.

**Figure 4 ijms-24-11809-f004:**
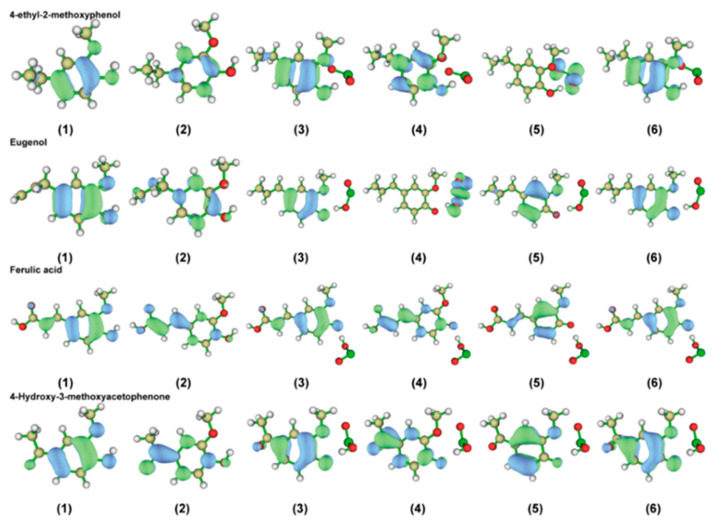
Homo–Lumo frontier orbit of phenolic lignin model compound and its transition state structure, (**1**) HOMO; (**2**) LUMO; (**3**) HOMO of alpha; (**4**) LUMO of alpha; (**5**) HOMO of bate; (**6**) LUMO of bate.

**Figure 5 ijms-24-11809-f005:**
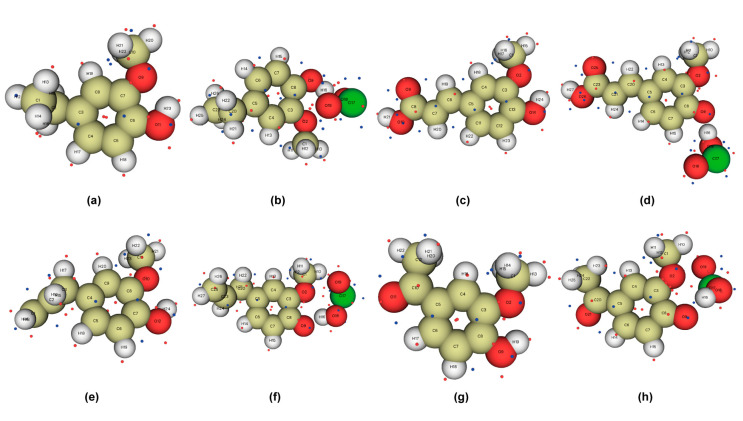
Distribution diagram of extreme minimum and partial minimum points of optimized molecular orbital weight double descriptor, (**a**) 4-E-ol; (**b**) E-ol; (**c**) FA; (**d**) APO; (**e**) 4-E-ol + ClO_2_; (**f**) E-ol + ClO_2_; (**g**) FA + ClO_2_; (**h**) APO + ClO_2_.

**Figure 6 ijms-24-11809-f006:**
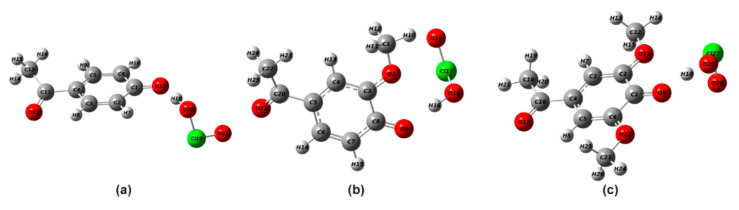
Geometric optimization of lignin phenolic model with different methoxy content, (**a**) 4-A-ol; (**b**) APO; (**c**) DHP.

**Figure 7 ijms-24-11809-f007:**
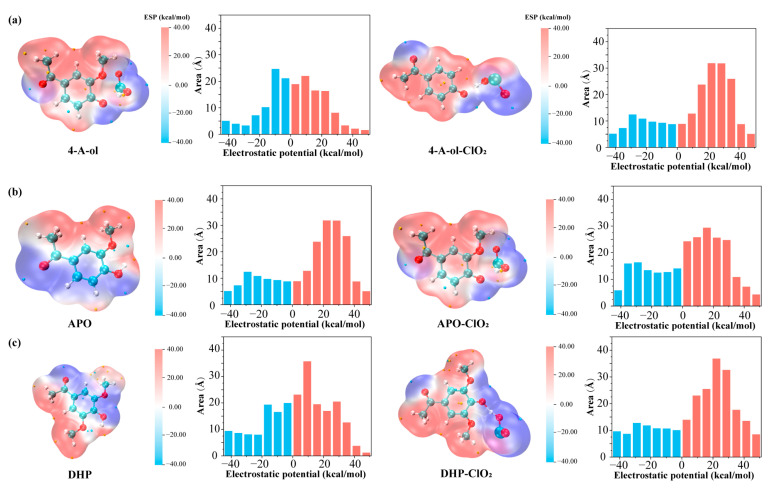
Quantitative distribution of electrostatic potential of lignin phenol model compounds with different methoxyl contents and lignin chlorine dioxide transition state structures, (**a**) 4-A-ol; (**b**) APO; (**c**) DHP.

**Figure 8 ijms-24-11809-f008:**
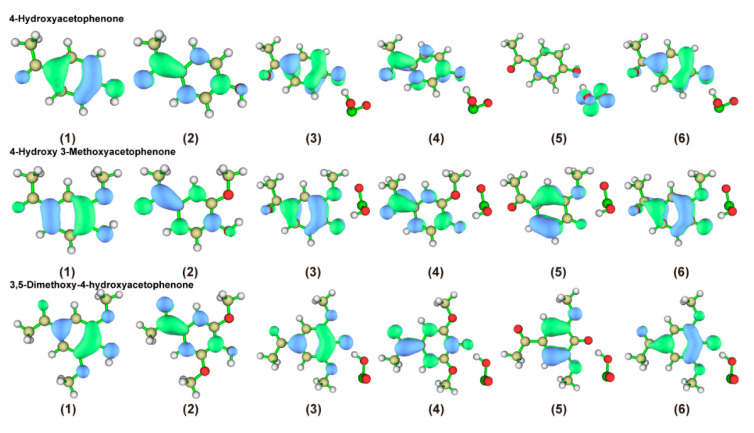
HOMO-LUMO frontier orbit of phenolic lignin model compound and its transition state structure (**1**) HOMO; (**2**) LUMO; (**3**) HOMO of alpha; (**4**) LUMO of alpha; (**5**) HOMO of bate; (**6**) LUMO of bate.

**Figure 9 ijms-24-11809-f009:**
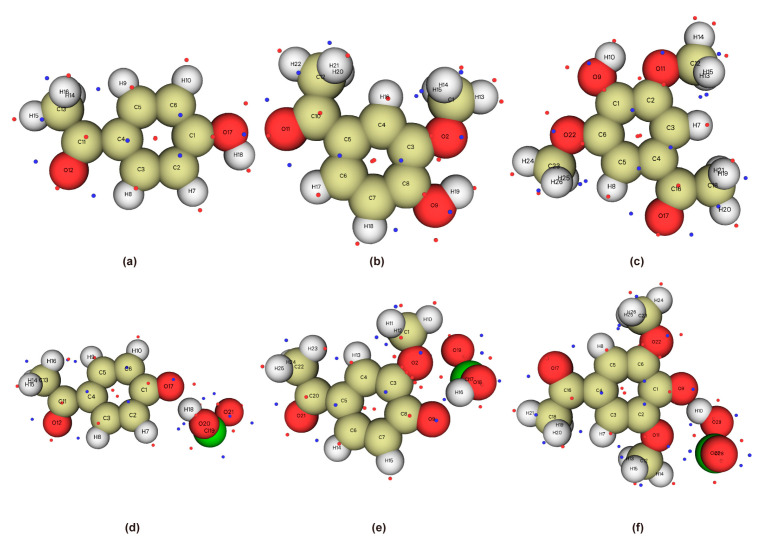
Distribution diagram of extreme minimum and partial minimum points of optimized molecular orbital weight double descriptor, (**a**) 4-ol; (**b**) APO; (**c**) DHP; (**d**) 4-ol + ClO_2_; (**e**) APO + ClO_2_; (**f**) DHP + ClO_2_.

**Table 1 ijms-24-11809-t001:** Chemical structure and names of phenolic lignin model compounds.

Lignin Model Compound	Scientific Name	Substituent Group		
		R_1_	R_2_	R_3_
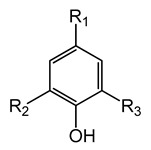	4-ethyl-2-methoxyphenol	-CH_2_CH_3_	H	-OCH_3_
	Eugenol	-CH_2_-CH-CH_2_	H	-OCH_3_
	Ferulic acid	-CH-CH-OOH	H	-OCH_3_
	4-acetylphenol	-O-CH_3_	H	H
	4-Hydroxy-3-methoxyacetophenone	-C-OH_3_	H	-COH_3_
	3,5-Dimethoxy-4-hydroxyacetophenone	-C-OH_3_	-OCH_3_	-OCH_3_

**Table 2 ijms-24-11809-t002:** The main bond length and bond angle information of phenolic lignin model substance.

Lignin Model Compound	Dissociation Energy E/(kJ·mol^−1^)	Bond Length (Å)	
	O-H	C-O	O-H
4-ethyl-2-methoxyphenol	84.38	1.3730	0.9679
Eugenol	86.09	1.3721	0.9680
Ferulic acid	87.35	1.3619	0.9688
4-acetylphenol	96.73	1.3607	0.9668
4-Hydroxy-3-methoxyacetophenone	90.36	1.3591	0.9690
3,5-Dimethoxy-4-hydroxyaceto- phenone	86.19	1.3583	0.9685

**Table 3 ijms-24-11809-t003:** C1 Hammett constants for phenolic lignin model species.

Lignin Model Compound	NBO Charge	Hammett Value
4-ethyl-2-methoxyphenol	−0.046	2.91
Eugenol	−0.049	2.86
Ferulic acid	−0.132	1.32
4-acetylphenol	−0.213	−0.17
4-Hydroxy-3-methoxyacetophenone	−0.189	0.27
3,5-Dimethoxy-4-hydroxyacetophenone	−0.166	0.70

**Table 4 ijms-24-11809-t004:** The main bond length and bond angle information of phenolic lignin model substance.

Lignin Model Compound	Bond Length (Å)			Angle (°)
	C-O	CO-H	ClO-H	O-H-O
4-E-ol	1.6243	1.0743	1.0429	174.7177
E-ol	1.6171	1.0096	1.0096	158.9603
FA	1.5691	1.0148	1.0148	179.6980
APO	1.6978	1.0296	0.9983	157.3091

**Table 5 ijms-24-11809-t005:** Frontline orbital energy information and HUMO-LUMO gap of phenolic lignin model substance and its transition state structure.

Compound	HOMO Energy (eV)	LUMO Energy (eV)		HUMO-LUMO Gap (eV)		
4-E-ol	−7.2558	0.4027		7.6586		
E-ol	−7.2618	0.3840		7.6458		
FA	−7.2889	−1.1542		6.1347		
APO	−7.6090	−0.7748		6.8342		
**Compound**	**HOMO Energy (eV)**	**LUMO Energy (eV)**	**Gap of** **Alpha** **(eV)**	**HOMO Energy (eV)**	**LUMO Energy** **(eV)**	**Gap of Bate** **(eV)**
4-E-ol + ClO_2_	−7.3411	−0.5542	6.7869	−8.4473	−3.1274	5.3199
E-ol + ClO_2_	−7.3431	−0.4371	6.9060	−8.4203	−3.1015	5.3188
FA + ClO_2_	−7.3508	−1.3546	5.9962	−8.5225	−3.5028	4.5027
APO + ClO_2_	−7.6401	−0.9664	6.6737	−8.6370	−3.5036	5.1334

**Table 6 ijms-24-11809-t006:** Reaction energy barrier formed by transition state of phenolic lignin model.

Index	4-E-ol	E-ol	FA	APO
E/au	−1110.8845	−1148.9597	−1298.2664	−1184.9299
ΔE/au	0.0311	0.0305	0.0328	0.0342
ΔE/kcal/mol	19.5153	19.1224	20.6040	21.4906

**Table 7 ijms-24-11809-t007:** The main bond length information of lignin phenolic model compounds with different methoxy content.

Lignin Model Compound	Bond Length (d/nm)			Angle (°)
	C-O	CO-H	ClO-H	O-H-O
4-A-ol	1.6352	1.0041	1.0041	165.9504
APO	1.6978	1.0296	0.9983	157.3091
DHP	1.2499	1.4509	1.0404	162.7035

**Table 8 ijms-24-11809-t008:** Frontline orbital energy information and HUMO-LUMO gap of phenolic lignin model substance and its transition state structure.

Compound	HOMO Energy (eV)	LUMO Energy (eV)		HUMO-LUMO Gap (eV)		
4-A-ol	−7.9882	−0.7469		7.2413		
APO	−7.6090	−0.7748		6.8342		
DHP	−7.5380	0.8059		6.7321		
**Compound**	**HOMO Energy (eV)**	**LUMO** **Energy (eV)**	**Gap of Alpha** **(eV)**	**HOMO Energy (eV)**	**LUMO Energy** **(eV)**	**Gap of** **Beta** **(eV)**
4-A-ol + ClO_2_	−8.2399	−0.9407	7.3993	−9.1822	−3.8501	5.3321
APO + ClO_2_	−7.6401	−0.96634	6.6737	−8.6370	−3.5036	5.1334
DHP + ClO_2_	−7.6151	−1.0629	6.5522	−8.1069	−3.5099	4.5970

**Table 9 ijms-24-11809-t009:** Reaction energy barrier formed by transition state of phenolic lignin model.

Index	4-ol	APO	DHP
E/au	−1070.6150	−1184.9299	−1297.6567
ΔE/au	0.0377	0.0342	0.0306
ΔE/kcal/mol	23.6598	21.4906	19.2268

## Data Availability

Not applicable.

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
