# Peer review of "Effect of Substituents on Molecular Reactivity during Lignin Oxidation by Chlorine Dioxide: A Density Functional Theory Study"

_ijms, 2023, doi:10.3390/ijms241411809_

Round 1
Reviewer 1 Report
In the present manuscript, Qin and Yao and coworkers quantum chemically analyze the chemical reactivity of ligning oxidation by chlorine dioxide. The authors focus the analysis on the transitions states of ligning model systems and ClO2, and the effect of different substituents. The study is performed with density functional theory and is based on the discussion of electrostatic potential, atomic charges, Fukui functions and reaction barriers. The interaction with ClO2 makes the polymer molecules to react via electrophilic substitution reactions. And this reactivity can be affected by the length of the substituent at position 1. Especially relevant is the role of methoxy substituent, which is clearly observed by means of the distributions of electrostatic potential. This manuscript is a good contribution to the understanding of the reactivity of this polymer, enclosing an extensive computational work, and for such I recommend publication in IJMS. My only suggestion is the citation of the following two works regarding the role of substituents on related chemical compounds by Alemán et al.: Physical Chemistry Chemical Physics 2012, 14, 10050-10062; Physical Chemistry Chemical Physics 2016, 18, 1265-1278.
Author Response
Dear Reviewer,
Thank you for your comments concerning our manuscript entitled "Effect of substituents on molecular reactivity during lignin oxidation by chlorine dioxide: a density functional theory study". We have studied the reviewer comments carefully and have made corrections which we hope could meet your requirements. See the track changes of the manuscript for all modifications.
Based on your comments, I make the following reply that needs to be revised.
- My only suggestion is the citation of the following two works regarding the role of substituents on related chemical compounds by Alemán et al.: Physical Chemistry Chemical Physics 2012, 14, 10050-10062; Physical Chemistry Chemical Physics 2016, 18, 1265-1278.
Thank you for your comments and for recognizing the work. Recommended references are cited in accordance with the opinion.
Reviewer 2 Report
The study provides a comprehensive analysis of the molecular properties and reactivity of lignin model species with ClO2, utilizing the m062x method of DFT at the 6-311g(d) basis group level.
The findings reveal that ClO2 activation renders lignin molecules highly susceptible to attack by electrophilic compounds, increasing the probability of electrophilic substitution reactions. The influence of substituents and methoxy groups on stability and reactivity is well-documented.
The study highlights the significance of different lignin structures, such as G-type and H-type, in relation to degradation and the formation of chlorite structures and quinone oxidation products. The impact of interlinked phenyl ether bonds linked to the S-type structure is also discussed.
It would also be beneficial to consider the broader implications of the study's findings and discuss how they could potentially be applied in practical applications, such as lignin valorization or lignin-based material synthesis.
Finally, minor revisions are needed throughout the manuscript to improve the organization and clarity of the presented information. Additionally, some technical terms and concepts could be better explained to make the study more accessible to a wider audience.
Overall, the study shows promise and has the potential to make valuable contributions to the field of lignin chemistry. Addressing the above points would significantly enhance the quality and impact of the research.
Author Response
Dear Reviewer,
Thank you for your comments concerning our manuscript entitled "Effect of substituents on molecular reactivity during lignin oxidation by chlorine dioxide: a density functional theory study". We have studied the reviewer comments carefully and have made corrections which we hope could meet your requirements. See the track changes of the manuscript for all modifications.
Based on your comments, I make the following reply that needs to be revised.
1、The study provides a comprehensive analysis of the molecular properties and reactivity of lignin model species with ClO2, utilizing the m062x method of DFT at the 6-311g(d) basis group level.
2、The findings reveal that ClO2 activation renders lignin molecules highly susceptible to attack by electrophilic compounds, increasing the probability of electrophilic substitution reactions. The influence of substituents and methoxy groups on stability and reactivity is well-documented.
3、The study highlights the significance of different lignin structures, such as G-type and H-type, in relation to degradation and the formation of chlorite structures and quinone oxidation products. The impact of interlinked phenyl ether bonds linked to the S-type structure is also discussed.
4、It would also be beneficial to consider the broader implications of the study's findings and discuss how they could potentially be applied in practical applications, such as lignin valorization or lignin-based material synthesis.
Thank you for your comments and for recognizing the work.
5、Finally, minor revisions are needed throughout the manuscript to improve the organization and clarity of the presented information. Additionally, some technical terms and concepts could be better explained to make the study more accessible to a wider audience.
It has been revised in the manuscript.
6、Overall, the study shows promise and has the potential to make valuable contributions to the field of lignin chemistry. Addressing the above points would significantly enhance the quality and impact of the research.
Thank you for your comments and for recognizing the work.